# Analysis of the Antimicrobial Market in Pakistan: Is It Really Necessary Such a Vast Offering of “Watch” Antimicrobials?

**DOI:** 10.3390/antibiotics8040189

**Published:** 2019-10-17

**Authors:** Farrukh Malik, Albert Figueras

**Affiliations:** 1Eu2P European Programme in Pharmacovigilance and Pharmacoepidemiology, Université de Bordeaux, 33076 Bordeaux, France; afs@icf.uab.cat; 2Fundació Institut Català de Farmacologia; Department of Pharmacology, Therapeutics, and Toxicology, Universitat Autònoma de Barcelona, 08035 Barcelona, Spain

**Keywords:** Antimicrobial market, irrational drug use, AWaRe, pharmacovigilance, antibiotic consumption, Pakistan

## Abstract

Understanding antimicrobials (AM) on offer in a pharmaceutical market, with a particular reference to drugs categorized as “Watch” active ingredients, is one of the important first steps to prevent their potentially inappropriate use. The March 2019 data of all AM containing registered brands and presentations currently being sold in the country for the J01 Anatomical Therapeutic Chemical (ATC) subgroup from IQVIA Pakistan was used. Each AM was categorized as “Access,” “Watch,” or “Reserve,” according to the WHO AWaRe classification. There were 59 single chemical entities and 14 combinations with 1869 brands and 4648 presentations. The WHO Essential Medicines List included 35 J01 AM while an additional 38 single and combination AM with 425 brands and 977 presentations were present in the country. Looking at the whole list of marketed AM, 8 of the 10 AM with more than 60 brands were classified as “Watch”, offering 962 brands and 2418 presentations. Most AM marketed in Pakistan, of which there are an excessive number of brands, belong to the “Watch” AWaRe category. The higher the number of brands, the higher the marketing pressure on prescribers and pharmacists will be which implies more potential confusion. One vital step to rationalize the use of AM lies in reviewing their market offer.

## 1. Introduction

In 1980, Lunde wrote that “there has been no proof of greater public health benefits with an infinite number of drugs” [1] and, to our knowledge, four decades later, nobody has proven the contrary. Additionally, in the case of antimicrobials (AM), an excess could be linked with higher resistance rates. Medicines marketed in a country are the result of cumulative marketing authorizations under different legal frameworks throughout history, some of them even before the modern evidence-based decisions era. So, as recommended by WHO, the first step to rationalize the use of medicines consists of an analysis of the pharmaceutical market [2].

Global AM market generated revenues of 42,335 million US$ in 2017, with projections of more than $50,000 million by 2025. Much of this growth is attributed to increased consumption of antibiotics in low and middle-income countries. Beta-lactam antibacterials and formulations, including beta-lactamase inhibitors, are and will be the most commonly prescribed antibiotics, followed by quinolones [3].

Over the next five years, Pakistan is expected to be among the top three developing countries with the fastest growth for medicine spending [4], which indicates a high consumption trend for antibiotics, with physicians having insufficient knowledge about their appropriate prescription [5], making it the third-highest consumer of antibiotics among lower and middle-income countries (LMIC) [6]. The risk of developing resistance significantly increases with the inappropriate use of antibiotics. Indeed, the growing Antimicrobial resistance (AMR) phenomenon has been dubbed as the “biggest threats to modern medicine” [7].

Antibiotics are categorized by WHO into three groups named “Access”, “Watch” and “Reserve”, a classification commonly known as ‘AWaRe” aimed at advising on how the different antibiotics should be available, selected, and prescribed. Thus, the “AWaRe” classification is also presented as a tool to help in the Global combat against antimicrobial resistance as each group specifies which antibiotics should be used as empiric (first or second line), cautiously or as a last recourse [8,9]. The global rise in AMR demonstrates that ill-coordinated efforts cannot help curb the challenge. When determining factors, apart from the biological predisposition of the organism to outgrow its threat, humanistic factors must be evaluated. The role of human decisions in terms of prescription, dispensing, prevention, self-use, and use in other sectors such as animals, are all important factors to study. Other macro-contributors to the phenomenon include the community, the environment, healthcare settings, and fragments and cracks in regulatory processes and systems to prevent antibiotic abuse [10]. The global antimicrobial stewardship program may serve as a platform to bring together these disparaging concerns and findings to make sense of the absurdity of overprescribing. As evident from sales data of Pakistan, the need to create multiple ‘me-too’ pharmaceutical products of the same active ingredient and even more so, its sales in such high numbers, is precisely the reason why international and local collaborations must take place under strict regulatory frameworks.

The healthcare system in Pakistan is one of the most challenged systems due to the least amount of gross domestic product (GDP) it receives to sustain itself, with less than 3% of GDP spent in 2016 [11]. The system is based on out of pocket payment, where services are given progressively according to the severity of the medical condition in primary, secondary, and tertiary care facilities [12]. The poor standards of care, and lack of resources, along with pressure from patients for medications from their care provider, all contribute towards antibiotic prescriptions.

Furthermore, there is a need to create and adhere to robust prescribing guidelines, which can help mitigate the irrational use of antibiotics [5].

The indicators developed by WHO have helped identify systematically the complex interplay of these factors and how, by keeping a close watch, improvements were achieved in reducing the usage of medicines in various regions. By setting surveillance methods, improvements were made in reducing the usage of medicines in various regions [13].

The National Strategic Framework for the Containment of Antimicrobial Resistance was adopted in 2017 as a national action plan to address the growing concern of antibiotic use and resistance. From the work carried out, multiple factors were identified as contributory to the challenge. Firstly, the number of antibiotics available in the Pakistani market is very high. Furthermore, there is highly prevalent self-medication behavior and up to 70% of patients are prescribed with antibiotics [14].

Antimicrobial stewardship should begin with a rational market offer of the available brands to be prescribed. Thus, the aim of the present study was to describe the antibiotics medicine offerings in Pakistan, with a particular reference to drugs that have been categorized as “Watch” active ingredients.

## 2. Results

Fifty-nine single chemical entities and 14 combinations belonged to the J01 ATC subgroup in Pakistan; there were 1869 brands containing one of these AM (median = 65 brands/AM; range = 23—684) and 4648 presentations (median = 130; range = 37–1981). The WHO Essential medicines List (WEML) included 35 AM under the ATC group J01. In Pakistan, an additional 38 single and combination AM were present with 425 brands and 977 presentations.

A high number of brands for antimicrobials belonged to the ATC subgroups J01M (Fluoroquinolones) and J01F (Macrolides), both of which are classified as the “Watch” category. Also, part of antimicrobials under ATC J01D (Cephalosporins) are classified as “Watch” (Figure 1).

The National Essential Medicines List of Pakistan (PEML) contains 31 AM from J01 class, while 6 AM (including one combination) present in the WEML were absent from the PEML, which belonged to the “Watch” or “Reserve” groups. These AMs included Cefuroxime, Cefuroxime Axetil, Cefotaxime, Imipenem+Cilastatin, Trimethoprim, & Fosfomycin with at least 103 brands (median 9.5, range = 2–62) and 238 presentations.

An in-depth analysis of the 38 additional AM, which were not considered in the WEML but available in the Pakistan market, showed that the number of brands per AM had a median value of 4.5 (range = 1–90). The top 20 AMs of these included, 13 AM are classified as “Watch” and two were classified as “Reserve” with 229 brands with a median value of 14 (range = 4–31) & 408 presentations (Table 1).

Looking at the whole list of marketed AM in Pakistan, 8 of the 10 AM with more than 60 brands in the market were classified as “Watch”, with 442 brands and 973 presentations of the different fluoroquinolones (ATC group J01M), 350 brands and 1060 presentations of third-generation cephalosporins (ATC subgroup J01DD) and 170 brands and 385 presentations of macrolides (ATC subgroup J01F).

It should be noted that for five “Reserve” group AM, which included two 4th generation cephalosporins (ATC subgroup J01DE), there were 57 brands available in the country.

## 3. Discussion

The main result of the present analysis of the antimicrobials available in the Pakistan market is apparently the excessive number of brands per active ingredient, which can be worrying in the particular case of some “Watch” and “Reserve” antimicrobials not included in the WHO Essential Medicines List. This is the case of lincomycin or the combination cefoperazone + sulbactam (with 31 different brand-names each), or the “Reserve” 4th generation cephalosporin cefepime, with 24 different brand names.

On the one hand, more available brand names of a single active ingredient mean more market pressure and, hence, more chances to be prescribed irrationally [15]. On the other hand, the high presence of “Watch” antibiotics in a market already saturated with countless combinations (rational and irrational) bears serious thought for policymakers and regulators [13].

Pakistan has become a fertile ground for promoting antibiotics, as the market regulations for medicines are weak, and the growth potential for generics (sold under brand names) is very high. There is an increasing need to identify the potential for harm or other interventions that can take place due to the high use of medicines in countries with lower socioeconomic status [16].

Interestingly, despite reports of antibiotics being regularly out of stock in many of the Pakistani public healthcare facilities, their sales are high in different pharmacies and regions. These primary healthcare facilities lack proper standard treatment guidelines for prescribers, with the result that almost 80% of the prescriptions in both outpatient and inpatient departments were found to be improper, and the most commonly prescribed antibiotics included fluoroquinolones, cephalosporins, and penicillins [17].

Due to misinformation amongst doctors as well as the general public, there is widespread use and buying behavior towards broad-spectrum antibiotics that are not followed through, leaving the antibiotic course in the middle [18], which poses a challenge in collating results, and may require more robust surveillance controls. Many solutions are being proposed to address concerns on medication safety, particularly the growing use of antibiotics. Alongside the need for systems that ensure the safety of medication use, there is a concurrent need for ensuring that guidelines developed must be aligned to international treatment standards.

The role of microbiologists should also be promoted as a way to improve the selection of the most appropriate antimicrobials for the patients, and this is especially relevant for “Reserve” AM [19].

In a recent effort to control antibiotic use, pharmacies in the capital, Islamabad, have been directed to keep a check on the number of antibiotics sold and that they should not be dispensed without prescription [20]. Despite this, dispensing without prescription is high in the country, and an excessive number of brands help to increase the irrationality of the decisions also at the pharmacy level.

The present research is a cross sectional analysis of the pharmaceutical market of antimicrobials in Pakistan based on data for the authorized medicinal products in the country. The main limitation is that the data source did not give any information on the products which were approved but had no recorded sales. Despite this, the study identified all antimicrobials available for prescription in the country; thus, it provided information on the wide range of different brands and presentations for a single active ingredient and combinations, and this was the main purpose of the analysis.

However, the findings of this study are not unique to Pakistan as there have been similar findings in other LMICs where only 50% of all the newly registered products were from the WHO’s essential medicines list [21]. Medicine market analyses such as the present one are quite simple to conduct and provide a general panorama of the products available in pharmacies of a given country. A careful observation of the results could show either irrational products marketed before the “evidence-based medicine era” or, as in the present case, an apparently unnecessary plethora of products with the same composition but different brand names which go beyond what could be considered a balanced assorted market. Countries could conduct these analyses as the first step in their antimicrobial stewardship programs.

## 4. Materials and Methods

The March 2019 data, which included sales units from the last 12 months for the J01 ATC subgroup (antimicrobials for systemic use) from IQVIA Pakistan, was used. Like many other LMICs, Pakistan lacks an updated official database for the authorized products with their sales and prescription figures. IQVIA Pakistan produces multiple types of retail data sets, but for the present research, Pakistan Pharmaceutical Index (PKPI) was accessible to study the pharmaceutical market as an indirect measure.

The file included all AM containing registered brands and presentations currently being sold in the country.

As a reference for the analysis, the 2019 edition of the World Health Organization Essential medicines List (WEML) [22] and the 2018 edition of the National Essential Medicines List of Pakistan (PEML) were used.

For the classification of the different antimicrobials, the well-recognized Anatomic Chemical Therapeutic (ATC) classification proposed by the WHO Collaborating Center for Drug Statistics Methodology in Oslo (Norway) was used, specifically, the antibacterials for systemic use group (J01) and its ten subgroups [23].

Additionally, each AM was classified according to the WHO AWaRe classification, as “Access,” “Watch,” and “Reserve” antimicrobials. “Access” antibiotics should be the first choice for the most common infections, while “Watch” includes most of the “highest-priority critically important antimicrobials recommended only for specific, limited indications. “Reserve” antibiotics should only be used as a last resort when all other antibiotics have failed [24].The inconsistent prescribing information for the same product in different countries have been observed which could have a detrimental impact on a patient [25]. This may also increase the potential for confusion when prescribed antibiotics medications are readily available over the counter without the need for prescriptions, further worsening the situation. As evidenced by sales data, the Pakistani market is currently selling the same generic product under multiple brand names. In this regard, packaging and critical information displayed on the packaging are important contributors to the quality of the information provided. It was found in a survey in the USA that over 33% of medication errors took place due to packaging issues or confusing information on the packaging [26].

This is just one example of how multiple branded generics under different names can cause confusion. In a market like Pakistan, where there is a lack of quality national resources for physicians to refer to for prescribing practices, the presence of multiple branded generics can lead to errors. Other contributors including marketing distractions, similar brand names for different products, misleading information on packaging, hidden drug fact labels, differing strengths and consumption methods, absence of visible warnings, and hidden ingredients may pose health risks [26]. In the case of antibiotics, physicians may be prone to make errors when prescribing doses, a matter that bears further investigation. 

## 5. Conclusions

In summary, Pakistan is a country facing high rates of antibiotic resistance; the excessive and apparently unnecessary high number of registered antibiotics available in the market, coupled with excessive use and prescription rates of these drugs, and lack of regulatory controls to counter irrational antibiotic prescription patterns could help to explain the present situation.

Most of the AM marketed in Pakistan belong to the “Watch” AWaRe category. With the marketing competitiveness of various brands at play and a higher number of brands available of a generic drug, more marketing pressure is likely to be applied to prescribers and pharmacists or vice versa.

So, there is a need to rationalize the use of AM by rigorously reviewing the AM market offer in light of international guidelines and standards. Furthermore, physicians must be educated and informed about the dangers of antibiotic resistance, encouraged to use antibiotics with caution, and made aware of their role in the prevention of this dangerous development.

## Figures and Tables

**Figure 1 antibiotics-08-00189-f001:**
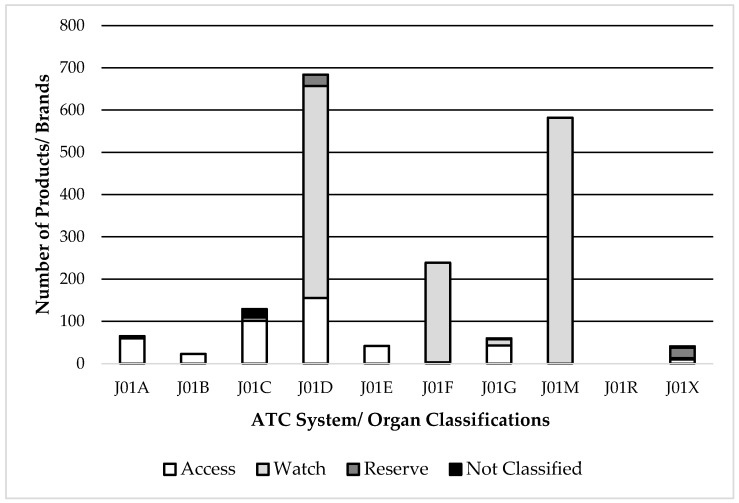
Distribution of the number of brands of all Antimicrobials for systemic use present in the country according to the Anatomical Therapeutic Chemical (ATC) classification, according to the WHO Access, Watch, reserve (AWaRe)classification. J01A = tetracyclines; J01B = amphenicols; J01C = beta-lactam antimicrobials, penicillins; J01D = other beta-lactam antimicrobials; J01E = sulphonamides and trimethoprim; J01F = macrolides, lincosamides and streptogramins; J01G = aminoglycoside antibacterials; J01M = quinolone antibacterials; J01R = combinations of antibacterials; J01X = other antibacterials.

**Table 1 antibiotics-08-00189-t001:** Top 20 antimicrobials not considered in the WHO Essential Medicines List but available in the Pakistan market according to the number of brands and presentations per active ingredient and classified according to the three WHO *AWaRe* categories.

*AWaRe* Category	ATC	Active Ingredient	Brands (*n*)	Presentations (*n*)
Access	J01DB09	cefradine	90	345
Access	J01DB05	cefradoxil	43	111
Watch	J01DD62	cefoperazone + sulbactam	31	62
Watch	J01FF02	lincomycin	31	51
Watch	J01MA15	gemifloxacin	24	24
Reserve	J01DE01	cefepime	24	44
Watch	J01FA06	roxithromycin	21	36
Watch	J01DC04	cefaclor	19	76
Watch	J01DD13	cefpodoxime proxetil	16	36
Access	J01AA06	oxytetracycline	14	17
Watch	J01MA06	norfloxacin	14	15
Watch	J01MA09	sparfloxacin	12	13
Not Classified	J01CR50	ampicillin + cloxacillin	11	31
Watch	J01GB04	kanamycin	9	15
Watch	J01MA03	pefloxacin	7	7
Watch	J01DD12	cefoperazone	6	10
Watch	J01GB01	tobramycin	6	9
Reserve	J01AA12	tigecycline	5	5
Not Classified	J01CR50	axoxicillin + flocoxacillin	5	11
Watch	J01MA16	gatifloxacin	4	5

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
