# Peer review of "Analysis of the Antimicrobial Market in Pakistan: Is It Really Necessary Such a Vast Offering of “Watch” Antimicrobials?"

_antibiotics, 2019, doi:10.3390/antibiotics8040189_

Round 1
Reviewer 1 Report
The manuscript analysed the number of brands and presentations for each type of antibiotic drugs available on the market in Pakistan using IQVIA data.
Overall the study made very limited contribution to both empiric knowledge of antibiotic prescribing practices in Pakistan/LMICs setting, theory of prescribing decision-making (from both health economic and health psychological perspective, and it did not contribute to research methodologies in antimicrobial resistance management and antimicrobial stewardship either.
The context of research was not explained, for example, who are prescribing antibiotics in Pakistan, how prescribing was regulated in different setting across health systems.
The data analysis results were poorly presented, for example, the only figure in the result sections has not axis labels. The WHO category of access, watch and reserve was not described in the manuscript, makes it extremely difficult for readers to interpret the study design and results. Also ATC category was not explained either (there are other types of antimicrobial in ATC classification system but not in J01 group).
The source datasets from IQVIA were not described clearly either. In fact if IQVIA data was used, the author could have use the actual consumption data for analysis instead of using only the data in branding - which led to very weak results of the analysis, saying that too many brands for the same type of drugs drove inappropriate prescribing. No previous research was referenced to at least provide some information in how such conclusion was drawn.
The manuscript provided a new angle in investigating inappropriate prescribing using market data.
Author Response
Response to Reviewer 1 Comments
General Comment: The manuscript analysed the number of brands and presentations for each type of antibiotic drugs available on the market in Pakistan using IQVIA data.
Point 1: English language and style: English language and style are fine/minor spell check required
Response 1: We have revised the text and improved typing, spelling, grammar and punctuation mistakes.
Point 2: Overall the study made a very limited contribution to both empiric knowledge of antibiotic prescribing practices in Pakistan/LMICs setting, theory of prescribing decision-making (from both health economic and health psychological perspective, and it did not contribute to research methodologies in antimicrobial resistance management and antimicrobial stewardship either.
Response 2: Dear reviewer, thank you for sharing your point of view. We have revised the whole manuscript taking into account these comments and have tried to clarify the primary purpose of the present study.
Inappropriate use of antimicrobials greatly contributes to the increase of antimicrobial resistance, so from the pharmacoepidemiological perspective, any analysis of the factors associated with the prescription, dispensing, self-medication, or the use of antimicrobials could provide useful information needed to design and implement interventions to improve drug utilization.
This could contain or reduce antimicrobial resistance; thus analyses of the antimicrobial consumption (and factors which influence that consumption) are part of a good stewardship program, as considered by WHO.
So, in our opinion, a critical analysis of the marketed antimicrobials, especially when the results obtained are as alarming as the ones found in Pakistan, with plenty of repeated molecules and so many generic brands, it is a necessary step to try to rationalize the market and thus, indirectly, try to improve the prescription. We have included these explanations both in the introductory paragraphs, as well as in the discussion.
Publishing this analysis could help health authorities to regulate effectively, and this can also be a model for other countries. Regarding the impact of such analyses of the medicines market (although it is not a new method), the obtained results are vital as the first step for a country interested in designing some actions to fight against antimicrobial resistance.
In fact, a similar kind of exercise is recommended by WHO, MSH, and other agencies to improve the use of medicines in general (beyond antimicrobials). Unfortunately, these critical analyses are not done in some countries (especially LMICs) for multiple reasons with various interests playing an important role.
Point 3: The context of research was not explained, for example, who are prescribing antibiotics in Pakistan, how prescribing was regulated in different setting across health systems.
Response 3: An effort has been made to include an explanation of the health system in Pakistan.
Point 4: The data analysis results were poorly presented; for example, the only figure in the result sections has not axis labels. The WHO category of access, watch, and reserve was not described in the manuscript, makes it extremely difficult for readers to interpret the study design and results. Also, ATC category was not explained either (there are other types of antimicrobial in the ATC classification system but not in J01 group)
Response 4:
The manuscript has been improved with explanations of the WHO ATC classification as well as the WHO AWaRe classification. Also, the omission of labels in the figure is corrected.
Point 5: The source datasets from IQVIA were not described clearly either. In fact, if IQVIA data was used, the author could have used the actual consumption data for analysis instead of using only the data in branding - which led to very weak results of the analysis, saying that too many brands for the same type of drugs drove inappropriate prescribing. No previous research was referenced to at least provide some information in how such a conclusion was drawn.
Response 5: The methods section has been improved, and why IQVIA was used as a proxy of the authorized medicines has now been explained more clearly than in comparison to the first version. The main reason is that the country (as it happens in many other LMICs) did not have an official list or an updated database with the authorized products. So, the only way to study the pharmaceutical market was an indirect measure, and the IQVIA sales data gave an overview of the total Pakistan retail market in their data set called Pakistan Pharmaceutical Index (PKPI).
Although the “excessive” number of brand names per active ingredient has never been defined, we have included a couple of references where authors are discussing the characteristics of the medicines market and the mind and behavior of prescribers.
Point 6: The manuscript provided a new angle in investigating inappropriate prescribing using market data.
Response 6: Thank you; this was precisely our idea when we designed this study. We wanted to look at the problem from another angle, especially in a data constraint environment and try to exemplify an approach which could be reproduced easily in other countries with similar characteristics.
Reviewer 2 Report
The manuscript does an overall good job describing the excessive number of brands and presentations of commonly used antibiotics in Pakistan. This excessive number of brands and presentations could lead to excessive marketing with resultant over use or over prescribing. This would have a significant impact on the development of antibiotic resistance. I have a few concerns and suggestions to improve the manuscript.
The WHO definition of Access, Watch, and Reserve antibiotics should be included in the Introduction of the manuscript to assist readers in interpreting the results. How does the findings of the number of brands and presentations of antibiotics in Pakistan compare to those of regional countries or internationally? If the findings are unique to Pakistan, what is the difference in medication use or dispensing that makes these findings unique to Pakistan? How do the findings for antibiotics compare to other classes of medications (e.g. cardiac or respiratory medications)? Are the findings unique to antibiotics?Author Response
Response to Reviewer 2 Comments
General Comment: The manuscript does an overall good job describing the excessive number of brands and presentations of commonly used antibiotics in Pakistan. This excessive number of brands and presentations could lead to excessive marketing with resultant overuse or overprescribing. This would have a significant impact on the development of antibiotic resistance. I have a few concerns and suggestions to improve the manuscript.
Response: Thank you for recognizing and appreciating as this was precisely our idea when we designed this study. We wanted to look at the problem from another angle, especially in a data constraint environment and try to exemplify an approach which could be reproduced easily in other countries with similar characteristics.
Point 1: English language and style: Moderate English changes required
Response 1: We have revised the text and improved typing, spelling, grammar and punctuation mistakes.
Point 2: The WHO definition of Access, Watch, and Reserve antibiotics should be included in the Introduction of the manuscript to assist readers in interpreting the results.
Response 2: The section has been improved with explanations of the WHO ATC classification as well as the WHO AWaRe classification. Also, the omission of labels in the figure is corrected.
Point 3: How does the findings of the number of brands and presentations of antibiotics in Pakistan compare to those of regional countries or internationally?
Response 3: Dear reviewer thank you for comments. It would have really interesting to compare the number of brands and presentations with the regional countries but finding an appropriate citable reference was difficult.
Point 4: If the findings are unique to Pakistan, what is the difference in medication use or dispensing that makes these findings unique to Pakistan?
Response 4: We feel that findings are not unique to Pakistan and there have been similar findings in other LMICs where only 50% of all the newly registered products were from the WHO’s essential medicines list [1] and we have included this point in the discussion of the revised version of the manuscript.
Point 5: How do the findings for antibiotics compare to other classes of medications (e.g. cardiac or respiratory medications)? Are the findings unique to antibiotics?
Response 5: It would have been really interesting to compare different ATC classes but with access to only the overview of all ATC classes and the only detailed info on Systemic Anti-infective (J Class) we couldn’t do the comparison in the present study. But since we know the ranking of the most sold (as mentioned below), we intend to design a study comparing the top 5 ATC classes.
|
ATC Code |
Organ Classification |
|
A |
Alimentary tract.& Metabolism |
|
J |
Systemic anti-infectives |
|
R |
Respiratory system |
|
N |
Nervous system |
|
M |
Musculo-skeletal system |
|
D |
Dermatologicals |
|
C |
Cardiovascular system |
References
Figueras, A., et al., Health needs, drug registration and control in less developed countries--the Peruvian case. Pharmacoepidemiol Drug Saf, 2002. 11(1): p. 63-4.

Reviewer 3 Report
This is a brief cross-sectional descriptions of antimicrobials brands being currently sold in Pakistan. Overall, the manuscript is easy to follow and read. I have only a few minor comments.
Minor comments
I suggest providing a brief description of the various AWaRe categories in the introduction, should this manuscript be occasionally read by scientists of other disciplines I found the discussion too long with respect to the descriptive cross-sectional data provided in the results. Overall, I think this manuscript should be shortened and could be better suited for a brief report. I would not call it an "in-depth" analysis, since it is limited to cross-sectional descriptive data.Author Response
"Please see the attachment."

Round 2
Reviewer 1 Report
I have finished my review and in general happy with the revised manuscript to be published. However I am a bit confused by the website interface - it looked like my comments in the first round will be wiped if I type in more comments. So I’d like to share with you my review comments directly in this email. After revision the author explained the rational better. However, I am still not convinced why there is no description in actual antibiotic consumption - essentially this is what IQVIA data is for - to measure drug sales. I have checked the author’s statement of data sources, they have obtain the IQVIA data from a pharmaceutical company in Pakistan. If the dataset is incomplete, please say so in the manuscript. Hope it helps.Author Response
Please see the attachment.

Reviewer 2 Report
All concerns from the original review have been adequately addressed in the revision
